# Facile Biogenic Synthesis and Characterization of Seven Metal-Based Nanoparticles Conjugated with Phytochemical Bioactives Using *Fragaria ananassa* Leaf Extract

**DOI:** 10.3390/molecules26103025

**Published:** 2021-05-19

**Authors:** Maryam Bayat, Meisam Zargar, Tamara Astarkhanova, Elena Pakina, Sergey Ladan, Marina Lyashko, Sergey I. Shkurkin

**Affiliations:** 1Department of Agrobiotechnology, Institute of Agriculture, RUDN University, 117198 Moscow, Russia; meisam.za_ir84@yahoo.com (M.Z.); tamara_ast@mail.ru (T.A.); e_pakina@yandex.ru (E.P.); nagvi@yandex.ru (M.L.); 2All-Russian Scientific and Research Institute of Agrochemistry, Federal State Budgetary Institution, 344006 Moscow, Russia; s.ldan@bk.ru (S.L.); s.shkurkin@sibac.ru (S.I.S.)

**Keywords:** metal-based nanoparticles, biosynthesis, nanoparticle characterization, strawberry leaf extract

## Abstract

In this investigation, for the first time, we used *Fragaria ananassa* (strawberry) leaf extract as a source of natural reducing, capping or stabilizing agents to develop an eco-friendly, cost-effective and safe process for the biosynthesis of metal-based nanoparticles including silver, copper, iron, zinc and magnesium oxide. Calcinated and non-calcinated zinc oxide nanoparticles also synthesized during a method different from our previous study. To confirm the successful formation of nanoparticles, different characterization techniques applied. UV-Vis spectroscopy, X-ray Diffraction (XRD) spectroscopy, Field Emission Scanning Electron Microscopy (FESEM) coupled with Energy Dispersive X-ray Spectroscopy (EDS), Photon Cross-Correlation Spectroscopy (PCCS) and Fourier Transformed Infrared Spectroscopy (FT-IR) were used to study the unique structure and properties of biosynthesized nanoparticles. The results show the successful formation of metal-based particles in the range of nanometer, confirmed by different characterization techniques. Finally, the presented approach has been demonstrated to be effective in the biosynthesis of metal and metal oxide nanoparticles.

## 1. Introduction

Metal-based nanoparticles (NPs) are used in many fields including agriculture, medicine, cosmetics, water treatment, electronics and catalysis, as their huge potential in nanotechnology is due to their ideal electrical, optical, magnetic and chemical properties [1,2]. Synthesis of metal-based NPs mainly mediates by physical or chemical methods. Physical methods require expensive equipment, high energy and large area allocation for equipment, while chemical ones require costly and toxic chemicals, which are not environmentally friendly. Also, supportive researches on plant-mediated biosynthesized nanoparticles indicate that biogenic nanomaterials are biocompatible and an effective therapeutic agent against bacterial, fungal infections, and cancer treatment [3]. The application of such NPs as nano-biopesticide and nano-fertilizer is more acceptable as they are benign for plants and cause less environmental contamination in comparison to conventional chemicals, as toxic chemicals may remain in the synthesized NPs and limit their application due to the toxicity of harmful residues, especially in the field of medicine, food industry and agriculture. To mitigate the problem of expensive equipment and toxic chemicals, green methods have been developed rapidly, presenting facile synthesis approaches [4,5,6].

Nowadays the use of biological systems such as plants and microorganisms has emerged as a novel issue in the biosynthesis of different nanoparticles. Plants are the most preferred source for biosynthesis of NPs because plant parts containing exclusive phytochemicals are involved in the reduction process of NPs biosynthesis. The rate of the biosynthesis in plant-mediated methods are more than microorganism-oriented methods, and the produced NPs are more stable and more diverse in shape and size. Plant extracts are widely applied in green synthesis of the metal-based NPs due to its the simplicity, ecofriendly and cost effectiveness for mass production of NPs and take much less time [5]. During the synthesis of NPs, extracted phytochemicals are added to metal salt solutions, reduce the metal ions and attach to the as-synthesized NPs acting as stabilizing or capping agents. These capping agents with no toxicity prevent NPs from aggregation. Moreover, as the kinetics of biological methods are slower than chemical methods, there results better control over the growth of NP crystals [7,8,9,10].

Different experimental conditions could lead to the production of NPs with the various characteristics such as color, size, shape and organic ligands binding to the surface of the synthesized NPs. These features affect potential applications and reproducibility of NPs formation. After the synthesis of NPs, the chemical composition and crystal structure of the synthesized NPs need to be investigated. Numerous experimental techniques are available to evaluate a variety of physical and chemical characteristics of biosynthesized nanoparticle samples such as size, shape, morphology, crystal structure and elemental composition [11,12]. Characterization approaches also help to conduct a reproducible synthesis approach and study the effect of different environments on characteristics of synthesized nanomaterials. Furthermore, an appropriate characterization of nanomaterials grants access of useful information to the researchers who investigate the application of NPs in different fields and also for regulatory bodies investigating NPs impact on healthcare and the environment [13]. Various techniques were used in this research for characterization of NPs.

Seven metal-based nanoparticles including C-ZnO, NC-ZnO, Zn, MgO, Ag, Cu and Fe have been biosynthesized in this research. There are many potential applications for such nanoparticles reported in the literature. ZnO and MgO NPs have antibacterial and anti-odor properties. These two nanoparticles are used as antibacterial ingredient in many products such as wood, food and cosmetics due to their increased ease in dispensability, optical transparence and smoothness [4]. There are only a few reports on applying biosynthesized Zn NPs. An effective antimicrobial activity of Zn NP against drug-resistant strains of community and hospital-acquired infections is reported, which suggests a new generation of antimicrobials for treatment, supplementary prophylaxis and prevention therapy [3]. In recent years, Ag NPs have attracted attentions because of their distinctive properties, such as high conductivity, chemical stability and mainly antibacterial activity [8]. Copper nanoparticles are widely used in the electrical industries because of their good conductivity and in chemistry as a lubricant and catalyst. Cu NPs also have been applied for water treatment and food processing [14]. Among metallic nanoparticles, Fe NPs have promising privileges that can combat environmental contamination. The most important application of Fe NPs is in environmental remediation. There is increasing interest in using nanoscale zero-valent iron in environmental remediation as nanoscale iron has more reactivity due to a large surface area to volume ratio [15].

The objective of the present report was the biosynthesis and characterization of seven NPs based on zinc, magnesium, silver, copper and iron salts. Strawberry leaf extract as an agro-waste economic material was used in this plant-mediated approach, for green synthesis of C-ZnO, NC-ZnO, MgO, Ag, Cu and Fe nanoparticles. To the best of our knowledge, it has not been reported in literature. Some of the potential applications of these biosynthesized metal-based NPs will be published in our future reports.

## 2. Results and Discussion

### 2.1. Biosynthesis of NPs

In this investigation, for the first time, the biosynthesis of NPs was carried out based on silver, copper, iron, zinc and magnesium salts employing an environmentally benign synthetic strategy, using strawberry leaf extract. Strawberry leaf extract contains minerals and biomolecules responsible for the biochemical reactions wherein biological molecules react with the metallic precursors leading to formation of the NPs (Figure 1).

### 2.2. Characterization of Biosynthesized NPs

In order to investigate the main characteristics of the biosynthesized nanoparticles, they were characterized using UV-Vis Spectroscopy, Field Emission Scanning Electron Microscopy (FESEM) coupled with Energy Dispersive X-ray Spectroscopy (EDS), Photon Cross-Correlation Spectroscopy (PCCS), Fourier Transformed Infra-red Spectroscopy (FT-IR) and X-ray Diffraction spectroscopy (XRD).

#### 2.2.1. UV-Visible Absorption Spectroscopy

UV–Vis absorption spectroscopy is the most widely used method for a preliminary analysis of metallic nanoparticle formation [16,17]. The optical absorption spectra of metal NPs and their intense color in water arise from surface plasmon resonances (SPR) phenomenon. SPR phenomenon is responsible for the color change of reaction mixture due to the excitation of SPR vibrations and reduction of metal ions (Figure 1). SPR vibrations are affected by different parameters such as temperature, pH, concentration, size and shape of synthesized NPs [18,19]. Therefore, the peak maximum position and sharpness changes proportional to the size and shape of NPs and move towards longer wavelengths, with the increase in particle size [20,21,22].

**Extract**. The UV-Vis spectrum of aqueous extract of strawberry leaf (Figure 2a) shows peak maximum at 250 nm due to the π/π* transitions strongly confirming the presence of aromatic compounds inside the plant extract [19].

**C-ZnO**. Synthesized C-ZnO NPs were subjected to UV-Vis spectrophotometer for confirming the formation of ZnO NPs at the first step (Figure 2b). The UV-Vis spectra of C-ZnO NPs indicated an absorption band at 355 nm, which is in the characteristic wavelength range of zinc oxide NPs and is consistent with previous studies [23,24].

**NC-ZnO**. The existence of biosynthesized NC-ZnO NPs also confirmed by the UV-Vis absorption spectroscopy. The spectra exhibited a strong absorption band at 345 nm presented in (Figure 2c) which is due to the surface plasmon resonance. The result is in good agreement with other reports [22,23,24].

**Zn**. In order to make sure that the Zn NPs were successfully synthesized, the UV-Vis spectroscopy was used. As shown in Figure 2d, there are two peaks of UV-Vis spectrum in the range of 200–800 nm. The peak maximum was observed at 270 nm, which is attributed to the Zn NPs.

**MgO**. UV–Vis spectra of pinkish colored biosynthesized MgO sample confirmed the formation of nanoparticles in the initial stage (Figure 2e). Various peaks were observed under UV region, at about 200 to 300 nm, which are in accordance with previous studies [17,25].

**Ag**. The UV-Vis measurements showed that the Ag NPs exhibit one peak in the UV-Vis region centered at 457 nm, which is in accordance with earlier published results (Figure 2f) [22,26].

**Cu**. The UV-Vis absorption spectra of Cu NPs showed an absorption band of around 270 to 370 nm (Figure 2g). The Cu NPs exhibit a dark brown color in aqueous solution due to the SPR excitation in UV-Vis spectrum depending upon the particle size. The similar absorption bands for Cu NPs have been reported in this range [27,28].

**Fe**. The UV-Vis spectra of Fe NPs (Figure 2h) showed prominent absorbance maximum at 270 nm and another low intense peak at 318nm, thereby confirming the production of Fe nanoparticles. These peaks were due to the excitation of SPR vibrations in the Fe NPs, which is consistent with the results of other researchers [29].

#### 2.2.2. Field Emission Scanning Electron Microscopic (FESEM) Images Analysis

The two most important parameters studied in the characterization of NPs are size and shape. Size and size distribution of the nanoparticles may affect other properties and possible applications of the NPs [11]. After confirming the biosynthesis of nanoparticles firstly by color change and then by UV-Vis spectroscopy, the morphology of nanoparticles was investigated by FESEM, which gives information about structural properties such as size, size distribution, shape, shape heterogeneity and aggregation of NPs. The resultant images are shown in Figure 3.

All of the biosynthesized NPs had sizes in the nanometer range, while some of them had nearly homogenous structure and yet others had heterogeneous structures (Table 1). Some nanoparticles were well separated from each other while most of them were present in agglomerated form. As a nanoparticle is defined a particle with at least one dimension in the range of nanometer, these FESEM results confirmed the nanostructure behavior of the synthesized particles [12]. The results are in agreement with previous reports, although with some slight differences due to chemical composition and synthesis conditions.

**C-ZnO**. Representative FESEM micrographs of the biosynthesized C-ZnO NPs are displayed in (Figure 3a). FESEM images show that the NPs have homogeneous spherical shape and the average particle size is 40 nm. It is also visible that the particles have agglomerated into larger clusters due to the attractive forces which bring them together into groups. The results are comparable with the results of 400–124 nm reported by Ogunyemi et al. [24].

**NC-ZnO**. FESEM image of the NC-ZnO NPs (Figure 3b) clearly demonstrated the presence of nano-scaled spherical shaped particles with average size of 25 nm which are agglomerated into larger clusters. It can be concluded from the FESEM studies that the calcinated product brings larger values of particle size (Table 1). The results are better than some previously published works and the particles sizes are considerably less than that (40–120 and 60–130 nm) [23].

**Zn**. FESEM images of biosynthesized Zn NPs (Figure 3c) show that the Zn NPs were sheets with irregular shape and the average size of 100 nm in diameter and 20 nm in thickness (Table 1). They also showed a small amount of agglomeration.

**MgO**. FESEM results of MgO NPs confirmed the nano range of the biosynthesized MgO nanoparticles with average size of 65 nm and quasi-spherical agglomerated nano-scaled particles (Figure 3d). Jeevanandam et al. reported similar results (18–80 nm) in their report [30].

**Ag**. Representative FESEM micrographs of the biosynthesized Ag NPs contains particles with uniform spherical shapes in the size diameter range of 50 nm and also depicts agglomerated Ag nano-crystals. These agglomerates easily dispersed in water with ultrasonic vibrations (Figure 3e). The obtained results are similar to the results of previous reports [27].

**Cu**. Based on the obtained images of FESEM (Figure 3f), the biosynthesized Cu NPs are in the nanometer range and it is evident that the morphology of crystals showed some non-uniform distribution. Their structure is in the form of sheets with average diameter of 180 nm and thickness of 30 nm (Table 1). Some nanoparticles were well separated from each other while most were present in agglomerated form. The results are in agreement with previous reports [31], although with few differences due to the synthesis condition.

**Fe**. FESEM micrographs of the synthesized Fe nanoparticles were shown in Figure 3g. The images represent nanoparticles of irregular shape like sheets with average diameter of 130 nm and thickness of 20 nm (Table 1) and the surface morphologies in the form of assemblies of nanoparticles.

#### 2.2.3. Energy Dispersive X-ray Spectroscopy (EDS) Analysis

EDX is a powerful technique allowing elemental analysis of the surface of solid samples such as synthesized nanoparticles. The EDS analysis shows the optical absorption peaks due to SPR of NPs. In EDS analysis the presence of each element is defined by the normalized weight percentage, which is the percentage in weight assuming that the chosen elements represent the total composition of the sample [32].

**C-ZnO**. The EDS pattern of C-ZnO NPs (Figure 4a) showed the peaks of zinc and oxygen elements. According to the elemental percentage analysis, the major element of zinc comprises more than 50% of total constituents, along with oxygen, which clearly confirms the formation of pure zinc oxide NPs. The EDS analysis shows that the samples contained pure zinc oxide nanoparticles which are in agreement with other researchers [23,24].

**NC-ZnO**. EDS pattern of non-calcinated zinc oxide NPs was shown in Figure 4b. The obtained elemental percentage of zinc and oxygen elements were 74.58% and 22.95%, respectively, and 2.47% potassium impurity. The EDS analysis are in accordance with Gupta et al. [33] and Dobrucka and Dugaszewska [34].

**Zn**. The EDS analysis profile of biosynthesized Zn NPs shows zinc (Zn), carbon (C), phosphorus (P) and oxygen (O) that confirms the involvement of plant phytochemical groups in reduction of metal ion and stability of the Zn NPs. EDS spectrum represents two peaks of zinc and oxygen (Figure 4c). The EDS pattern of Zn NPs is in agreement with Hosseini Koupaei et al. [35].

**MgO**. EDS spectrum reveals the presence of Mg and O elements in biosynthesized MgO NPs (Figure 4d). It is released from the EDS analysis that the MgO NPs are composed of Mg, O and a small amount of Ca as impurity. The molecular ratio of Mg:O was found to be nearly 1:1, which shows the purity of the NPs. Mentioned result was in good agreement with previous reports [25,36,37].

**Ag**. The EDS spectrum presented in Figure 4e shows the three characteristic peaks of silver around 3 keV due to SPR. According to the elemental profile, silver abundance is about 98.05% normalized mass percent. The other element is Cl with 1.95% relative abundance, which is related to the impurities of the precursor. This result was in good agreement with Masoud Hussein et al. [38].

**Cu**. The EDS spectra of biosynthesized Cu NPs is shown in Figure 4f. It is clear from the spectra that the EDS pattern confirmed successful formation of Cu NPs, as the EDS peak positions were consistent with copper. The presence of other elements such as oxygen (O) and carbon (C) in the Cu NPs is related to the biomolecules of the extract attached to the surface of Cu NPs [39]. These findings are in close agreement with the previous studies [24], but with slight difference due to variations in synthesis condition.

**Fe**. Figure 4g shows the elemental profile of biosynthesized Fe NPs, which shows an intense signal indicating the presence of elemental Fe in examined samples. The elemental analysis of the Fe NPs revealed strong signals of O (48.75%), C (36.85%) and Fe (12.43%) along with weak signals of P (1.97%), as the impurities come from extract biomolecules. This result was in accordance with Sylvia Devi et al. [40].

#### 2.2.4. Photon Cross Correlation Spectroscopy (PCCS) Analysis

PCCS analysis gives the radius of a particle and an estimation of the average hydrodynamic particle size. The average hydrated particle size distribution of the NPs synthesized by employing the present novel green strategy is presented in Figure 5, and the details are summarized in Table 2. As an example, the average hydrated particle size was 133 nm for Ag NPs. The observed results obviously revealed that the size obtained from PCCS was not correlated with FESEM observations because these results are related to hydrated aggregations of NPs, not single ones. The size analysis suggests that Ag NPs with the average size of 133 nm were comprised of nanocrystals with an average size of about 50 nm. The biosynthesized NPs existed as agglomeration of small individual nano sized particles, as there is a considerable tendency to form non-uniform sized and shaped agglomerates [22].

The polydispersity index (PDI) indicates the heterogeneity of a sample based on size and can occur due to size distribution in a sample or agglomeration of the sample. As Equation (1):(PDI) = the square of the (standard deviation/mean diameter)(1)

PDI value ranges from 0–1 and 0–0.08 indicate good monodispersity of the sample, 0.08 to 0.7 are mid-range and values more than 0.7 show very broad distribution of particle sizes [26]. The values of PDI for the biosynthesized NPs are presented in Table 2.

#### 2.2.5. Fourier-Transform Infrared Spectroscopy (FT-IR)

The FT-IR spectra were recorded in the range of 400–4000 cm^−1^ to recognize the functional organic groups or biomolecule residues along with biosynthesized NPs, which may come along via strawberry leaf extract due to the interaction of NPs with capping agents. The obtained FT-IR spectra confirmed the interaction of biomolecules with synthesized NPs. Table 3 summarizes the absorption bands of all spectra obtained from the extract and biosynthesized NPs.

**Extract**. The FT-IR spectrum of the strawberry leaf extract (Figure 6a) indicates absorption bands at 3927 cm^−1^, 3539 cm^−1^, 3392 cm^−1^, 2028 cm^−1^, 1624 cm^−1^, 1084 cm^−1^, 770 cm^−1^, 621 cm^−1^, 474 cm^−1^ and 423 cm^−1^ (Table 3). Intense peaks at 3539 cm^−1^ and 3392 cm^−1^ may be due to the N–H stretching frequency of peptide linkages or O–H stretching vibrations. The peak at 2028 cm^−1^ was assigned to the C-H aromatic. The band at 1624 cm^−1^ most probably refers to C=O or C=C aromatic ring stretching vibrations and 1084 cm^−1^ indicates the frequencies of C-O or C-N amine. At 770 cm^−1^, a weak peak appears which may be related to the alkane, alkene or aromatics C-H band or C-S thiol or thioether [41]. The results indicate the presence of different biomolecules in strawberry leaf extract and most probably polyphenols, carboxylic acid, amino acids and proteins [41,42]. In the present study, the interaction of NPs with biomolecules of the extract showed intense peaks and relative shift in position and intensity distribution (in comparison to extract) which were confirmed with FT-IR.

**C-ZnO**. Figure 6b shows the FT-IR spectrum of the calcinated ZnO NPs. Absorption bands occurred at 418 cm^−1^, 481 cm^−1^, 623 cm^−1^, 1079 cm^−1^, 1424 cm^−1^, 1513 cm^−1^, 1621 cm^−1^, 2030 cm^−1^, 3238 cm^−1^, 3415 cm^−1^, 3480 cm^−1^, 3553 cm^−1^ (Table 3). The intense broad bands at 3553 cm^−1^, 3480 cm^−1^ and 3415 cm^−1^ were assigned to O–H stretching of flavonoids, polyphenols and C–O and N–H groups on the surface of ZnO NPs [23]. 3238 cm^−1^ indicates the frequencies of aromatic or alkene C–H groups [33]. The intense peak at 2030 cm^−1^ indicates the C–H of aromatic ring. The peak at 1621 cm^−1^ corresponding to C=C stretch in the aromatic ring or C=O group. Peaks at 1513 and 1424 cm^−1^ refer to N-O nitro, C = C or amine N–H stretching of the aromatic compound which is observed after calcination process. Bands at 1079 cm^−1^ indicates the stretching vibration related to C–N of an amine or C–O. Absence of 1395 cm^−1^ sharp peak (in comparison with NC-ZnO in the following section) due to antisymmetric stretching vibration of O–H in the Zn(OH)_2_. These findings show the presence of organic functional groups absorbed on the surface of C-ZnO nanostructures, which may present in strawberry leaf extract. The results are in accordance with previous reports [23,24,33].

**NC-ZnO**. According to FT-IR spectrum of the C-ZnO NPs (Figure 6c), the bands are at 3553 cm^−1^, 3475 cm^−1^, 3411 cm^−1^, 3237 cm^−1^, 2028 cm^−1^, 1620 cm^−1^, 1395 cm^−1^, 1069 cm^−1^, 623 cm^−1^, 482 cm^−1^, and 418 cm^−1^. The band at 3553 cm^−1^ 3475 cm^−1^ and 3411 cm^−1^ are due to stretching vibrations of O–H groups or N-H amine group (Table 3). The C–H stretch in alkene showed broad absorption bands at 3237 cm^−1^. The absorption band at 2028 cm^−1^ is characteristic of the aromatic C–H stretching vibrations. The small peak located at 1620 cm^−1^ could be assigned to the C=O stretching in the carboxyl or C=C stretch in aromatic ring. The sharp band at 1395 cm^−1^ refers to the N=O vibrations of nitro compound which is formed due to the calcination process. The peak at 1069 cm^−1^ is due to C–O stretching in amino acid and vibration of (NH)–C=O group. It may conclude that the biomolecules which are responsible for the reduction of metal ions remain on the surface of the ZnO NPs for their stabilization.

**Zn**. The FT-IR spectra of Zn NPs (Figure 6d) displays a number of absorption peaks explained in Table 3. The broad peaks of 3553 cm^−1^, 3479 cm^−1^ and 3416 cm^−1^ could be attributed to N-H stretching of the aliphatic primary amine, or O–H stretching alcoholic or phenolic groups or different carboxylic acids of the extract. The C–H stretch in alkene showed broad absorption bands at 3239 cm^−1^. The absorption band at 2039 cm^−1^ is due to the aromatic C–H stretching vibrations. The sharp peak located at 1620 cm^−1^ could be attributed to the C=C ring stretching vibrations in polyphenols or asymmetric stretching vibrations of COOH, and that around 1148 cm^−1^ corresponded to the C–N amine or C–O stretching frequency [43]. FT-IR analysis of this compound suggested the presence of organic components from strawberry leaf extract [34,35].

**MgO**. The FT-IR spectrum of MgO NPs is shown in Figure 6e and summarized in Table 3. The broad absorption peaks at 3554, 3482, 3417 cm^−1^ are due to the stretching of N-H stretching of the aliphatic primary amine, or O–H stretching alcoholic or phenolic groups or different carboxylic acids of the extract. The peak at 3233 cm^−1^ may related to the C–H stretch in alkene. The absorption at 2362 most probably refers to the CO_2_ molecules released in the calcination process. The absorption band at 1996 cm^−1^ is for aromatic C–H stretching vibrations. The peak at 1621 cm^−1^ could be due to the C=C ring stretching in polyphenols or stretching vibration of carboxyl group. Peak at 1549 cm^−1^ corresponded to N-O nitro compound, which is produced during calcination process. At 1078 cm^−1^ an absorption band occurred which could be related to the C–N amine or C–O stretching frequency [37].

**Ag**. The FT-IR spectra of Ag NPs (Figure 6f, Table 3) confirms the presence of residual capping agent with the biosynthesized Ag NPs. The bands observed at 3552, 3479, 3416 cm^−1^ in the spectra correspond to the amine N-H and O–H stretching vibration indicating the presence of alcohols, phenolic-related species and proteins. The band detected at 3239 cm^−1^ is related to the C–H stretching of alkene compound. The peak at 2090 cm^−1^ can be assigned to the C–H stretching of aromatic compound and the strong band that appeared at 1622 cm^−1^ corresponded to the C=O or C=C aromatic ring vibration. The band at 1151 cm^−1^ could be assigned to the C-O or C-N amine, suggesting the presence of proteins. The band at around 621 cm^−1^ is due to the C-H alkyne bond or O-H out of plane bending vibration. Such detected functional groups have been also reported in previous investigations [38,43]. It can be concluded from the FT-IR observations that the surface of the Ag NPs is covered by organic species of the extract.

**Cu**. FT-IR spectra of Cu NPs presented in Figure 6g and Table 3 shows strong absorption bands at 3553, 3479 and 3416 cm^−1^ related to the amine N-H and O–H stretching vibrations indicating the presence of alcohol, phenolic-related species and protein. The peak at 3239 cm^−1^ is attributed to the C–H stretching of alkene compound and 2923 cm^−1^ may refer to the C-H, secondary amine or amine salt. The peak at 2039 cm^−1^ is related to the C-H aromatic vibrations and 1622 cm^−1^ is attributed to binding of C=O or C=C aromatic ring to metal ions. The sharp band at 1327 cm^−1^ refers to the S=O vibration of the sulfate substrate remaining from copper sulfate using in synthesis of Cu NPs. Another peak at 1151 cm^−1^ may refer to the C-O or C-N amine functional group. The peak at 621 cm^−1^ is related to the C-H alkyne bond or O-H out of plane bending. These findings are in agreement with previous reports [27].

**Fe**. The FT-IR spectrum of the synthesized Fe NPs is represented in Figure 6h and Table 3. Intense bands at 3523, 3479 and 3416 cm^−1^ refer to the N–H or O–H stretching vibrations and the band at 3239 cm^−1^ may related to the C-H alkene. The peak at 2922 cm^−1^ is due to the C–H and CH_2_ vibration of aliphatic hydrocarbons, secondary amine or amine salt. At 2039 cm^−1^ the peak may arise from C-H aromatic band, and at 1622 cm^−1^ may be due to the C=O or C=C aromatic ring stretching vibrations. The absorption band at 1144 cm^−1^ may related to the C-O or C-N amine, and finally absorption bands at around 622 cm^−1^ refer to C-H alkyne bond or O-H bending. Comparison of the IR spectrum of extract with the Fe NPs, shows a band shift towards higher frequencies from 3392 to 3416 cm^−1^ (O–H stretching vibrations), 2028 to 2039 cm^−1^ (aromatic C–H) and 1084 to 1144 cm^−1^ (C-OH bending) along with the presence of 1622 cm^−1^ (C=C stretching vibrations), which indicates the possible attachment of the polyphenolic compounds of the extract in the reduction step through the hydroxyl group and attaching to the Fe NPs through the carbonyl groups.

#### 2.2.6. X-ray Diffraction (XRD) Analysis

XRD is a useful and valuable characterization technique in obtaining information about the atomic structure of materials and indicating formation of NPs. An advantage of the XRD technique is that it results in statistically representative and volume-averaged values. The chemical composition of the nanoparticles could be determined by comparing the obtained peak patterns with the reference patterns available from the International Centre for Diffraction Data (ICDD) database. However, it is not suitable for amorphous materials and for particles with a size below 3 nm, as XRD peaks are too broad in such cases [11].

Synthesized NPs were subjected to X-Ray diffraction studies, confirm the presence of the NPs, obtain the crystallinity or amorphic nature and to detect possible impurities. XRD also used to calculate the average crystallite size of the NPs by Scherrer’s equation, using the broadening of the most intense peak of an XRD measurement for each sample [11,44].

**C-ZnO**. Figure 7a shows the XRD pattern of calcinated zinc oxide NPs. Number of Bragg reflections for biosynthesized calcinated ZnO NPs appears at 2θ = 31.77° (100), 34.44° (002), 36.26° (101), 47.55° (102), 56.61° (110), 62.89° (103), 66.39° (200), 67.97° (112), 69.10° (201).72.61° (004), 76.99° (202), 81.43° (104) and 88.65° (203).

The planes show a good agreement with the COD file (COD database code: 96-900-4181) [45], which elucidates the hexagonal structure and corresponds to pure zinc oxide nanocrystals with no impurity patterns.

Using Scherrer’s equation [16], the average particle size of biosynthesized NPs calculated to be 25.3 nm for C-ZnO NPs, which matches the average diameter of particles determined by FESEM. This finding suggests that the NPs are mainly single crystals rather than polycrystalline [16]. Diffraction peaks of other impurities were not detected. These outcomes are similar to previous works [23,25,33].

**NC-ZnO**. According to the Figure 7b, peaks for biosynthesized non-calcinated ZnO NPs appears at 2θ = 31.73° (100), 34.38° (002), 36.21° (101), 47.48° (102), 56.53° (110), 62.77° (103), 66.30° (200), 67.86° (112), 69.00° (201), 72.46° (004), 76.86° (202), 81.27° (104) and 89.49° (203) respectively. The planes show a good agreement with the COD file (COD database code: 01-076-0704, calculated from ICSD using POWD-12++, 1997) [46], which shows the hexagonal structure, corresponded to pure zinc oxide NPs without any impurity patterns.

For the NC-ZnO NPs, the calculated average particle size using Scherrer’s equation was 27.46 nm. These findings are very close to the particles sizes attained from FESEM analysis. Previous studies have reported similar results [34,43].

**Zn**. The XRD pattern of the synthesized Zn NPs is provided in Figure 7c. The pattern shows no distinctive diffraction peaks, suggesting that the synthesized Zn NPs are largely amorphous in nature rather than crystalline, or the size of the individual NPs are smaller than detection limit of the XRD. Also, it could be suggested that the amount of the crystals was lower than the XRD detection level. As there is no diffraction peak, the particle size cannot be calculated using Scherrer’s equation.

**MgO**. Figure 7d shows the XRD patterns of biosynthesized MgO NPs. The wide angle XRD pattern indicates the presence of cubic structure of MgO NPs determined with diffraction peaks at 2θ values are 37.04° (111), 43.04° (200), 62.49° (220), 47.93° (311) and 78.89° (222). No subsequent peaks of Mg(OH)_2_, Mg or other impurities were detected in the XRD pattern of MgO NPs, showing the high purity of the synthesized nanocrystals. The obtained data was matched with reported COD data card no: 01-087-0653, calculated from ICSD using POWD-12++, (1997) [47]. The average grain size 69.6 nm was calculated using Scherrer’s equation which is in close agreement with the size of 65 nm obtained from FESEM. The broad peaks indicate the decreasing in crystallinity, which inwardly suggests the formation of smaller particles size. The obtained results are also consistent with previous studies [48].

**Ag**. The XRD pattern of biosynthesized Ag NPs is in Figure 7e. The XRD of the Ag NPs between 2*θ* values of 20° and 90° exhibits crystalline nature and is in agreement with earlier published data showing peaks of silver NPs [20]. Bragg’s diffraction peaks for biosynthesized Ag NPs are observed at 2*θ* = 38.11° (111), 44.30° (200), 64.45° (220), 77.40° (311), and 81.55° (222). The sharp diffraction peaks confirm the crystalline nature of the sample. According to the COD data card (no: 01-087-0717), the standard diffraction peaks represent the face centered cubic (fcc) structure of Ag NPs, calculated from ICSD using POWD-12++, (1997) [49]. Diffraction peaks of other impurities were not observed, depicting that the peaks belonged to the pure Ag NPs.

The mean grain size of the Ag NPs calculated using Scherer’s formula was 40.9 nm, which is close to the obtained size of 50 nm resulted from FESEM analysis.

**Cu**. Figure 7f shows the XRD pattern of Cu NPs, which showed no peaks assign to crystalline structure. According to the results, biosynthesized Cu NPs are amorphous and/or the particle size is too small. Also it could be concluded that the amount of the nanocrystals was lower than the XRD detection level.

It can be seen that the XRD patterns consist of small shape peak between diffraction angles 20° and 30°, which may indicate some crystalline phase mixed with the primary amorphous structures. The XRD pattern is in agreement with previous reports, but with little differences [50].

**Fe**. XRD pattern of biosynthesized Fe NPs is shown in Figure 7g. The pattern exhibits no discernible diffraction peak corresponding to the crystalline nature of Fe NPs. Results assigned to amorphous particles and show no crystalline phases formed. It also may be due to the fact that the percentage of amorphous particles dominated the amount of crystalline particles. Another probability is that the size of NPs is small and cannot be detected by XRD.

Formation of amorphous Fe NPs using leaf extract of different plants was also reported in other published works [51].

## 3. Materials and Methods

**Extract preparation**. The extract prepared in a method as we mentioned in our previous report [1], by mixing 10 g powder of dried leaves with 100 mL distilled water, boiling, centrifuging and filtration of the obtained extract.

### 3.1. Biosynthesis of Nanoparticles

**ZnO**. In this investigation, for synthesis of ZnO NPs we tried a different method from our previous work [1]. Zinc acetate dihydrate, Zn(CH_2_COO)_2_·2H_2_O and strawberry leaf extract applied for the synthesis of calcinated and noncalcinated ZnO nanoparticles. 10 mL of the extract added dropwise to the 100 mL 0.01 M solution of the zinc salt, which was stirring with magnetic stirrer and heated to 90 °C. As the reaction mixture became sturdy and brown showing the formation of NPs and the process followed by addition of 2 M potassium hydroxide solution drop by drop to the pH of 11. Then the reactant was centrifuged at 10,000 rpm for 25 min and washed several times until the supernatant became colorless. The obtained brown-color gel was dried at room temperature and ground very well to give a fine powder of non-calcinated ZnO (NC-ZnO) NPs. For production of calcinated ZnO NPs, the obtained NC-ZnO NPs was placed in furnace at 500 °C for 4 h. Finally, the white powder of calcinated ZnO (C-ZnO) was ground well and stored for further use.

**Zn**. Zinc acetate dihydrate, Zn(CH_2_COO)_2_·2H_2_O salt used for the synthesis of the Zn NPs. 10 mL of the leaf extract was added drop by drop to the100 mL 0.01 M solution of the zinc salt under heating (90 °C) and stirring with magnetic stirrer. The color change of mixture into the cloudy brown indicated the formation of Zn NPs and then it was centrifuged at 10,000 rpm for 25 min and washed several times. The brown-color pelletlets dried at room temperature, ground and kept in a sealed container for further use.

**MgO**. Magnesium sulfate heptahydrate (MgSO_4_·7H_2_O) was the precursor for the synthesis of MgO nanoparticles. 100 mL of the prepared 0.01 M solution of magnesium sulfate in distilled water was heated to 90 °C. 10 mL of the leaf extract was added drop by drop under continuous stirring and after that continued by addition of 2 M potassium hydroxide solution dropwise to the pH of 11. Addition of KOH resulted in formation of precipitation and turbidity of the solution, confirming the formation of NPs. The obtained solution centrifuged at 10,000 rpm for 25 min and washed several times. Resultant viscous dark brown gel was dried at room temperature and then ground perfectly, calcinated in furnace at 500 °C for 4 h to becomes a pinkish white powder of MgO NPs, then ground another time.

**Ag**. Silver nitrate (AgNO_3_) used as the precursor for producing Ag NPs. Silver nanoparticles produced by reducing silver ions using strawberry leaf extract. 100 mL of AgNO_3_ 0.01 M solution was subjected to continuous stirring, heating to 90 °C and then addition of leaf extract drop by drop. By addition of the extract the color of the reaction mixture was changed to dark gray and became opaque which indicated the formation of Ag NPs. The solution was kept overnight at room temperature and then was centrifuged at 10,000 rpm for 25 min and washed several times. Ag NPs was dried at room temperature and then ground to obtain very fine NPs.

**Cu**. The precursor of the synthesis of Cu NPs was copper (II) sulfate pentahydrate (CuSO_4_·5H_2_O) salt. 10 mL of the leaf extract added dropwise to the 100 mL 0.01 M solution of the copper salt under the continuous magnetic stirring. A brown cloud of the Cu NPs appeared by the addition of extract solution. The resultant solution remained at room temperature overnight for completing the precipitation process. To collect the sample, the solution was centrifuged at 10,000 rpm for 25 min, washed several times, precipitant dried at room temperature and ground perfectly.

**Fe**. Iron (III) chloride hexahydrate (FeCl_3_·6H_2_O) salt was used for the synthesis of the Fe NPs. 10 mL of the leaf extract was added dropwise to the 100 mL stirring 0.01 M solution of the iron salt. The color of the reaction mixture changed immediately from yellow to intense black, indicating the formation of Fe NPs. Then the mixture centrifuged at 10,000 rpm for 25 min and washed several times until the supernatant became colorless. The black-color precipitant was dried at room temperature and finally ground to result in a uniform and smooth powder.

### 3.2. Characterization of Biosynthesized NPs

UV-visible absorption spectroscopy. A UV-Vis spectrophotometer model Varian Cary 50 (Palo Alto, CA, USA) was employed in the range of 200 to 800 nm to confirm the production of NPs and explore the absorption spectra of NP suspensions. 1 mg/mL aqueous suspensions of the NPs were prepared freshly before the spectroscopy and sonicated for 20 min. Distilled water was applied as the reference for all measurements.

Field emission scanning electron microscopy (FESEM) with energy dispersive X-ray spectroscopy (EDS). FESEM-EDS analysis was performed to obtain information about size, size distribution, shape heterogeneity and aggregation of the synthesized NPs and their elemental composition. TESCAN MIRA3 (Brno-Kohoutovice, Czech Republic) instrument was applied for FESEM-EDS analysis. In order to prepare samples for taking the FESEM images, samples were coated by a very thin layer of gold to improve the sample surface conductivity, prevent charge accumulation and obtain a better contrast. Data about mean size was obtained by measuring the particle size of about 100 nanoparticles. For analysis of SEM and FESEM images the software “imageJ” (1.53 e, Wayne Rasband and contributors, national institutes of health, Madison, WI, USA) was used.

Photon cross correlation spectroscopy (PCCS). Hydrodynamic particle size distribution of the synthesized NPs was measured by a photon cross correlation spectroscopy (PCCS) using a Nanophox particle size analyzer (Sympatec, Clausthal-Zellerfeld, Germany). Prior to measurement, 1 mg/mL aqueous suspensions of the nanoparticle were prepared and sonicated for 20 min at 25 °C. The measurements were performed at room temperature and repeated five times for each sample.

Fourier-transform infrared spectroscopy (FT-IR). To study peak patterns of leaf extract, and the bonding patterns of capping agents present in the extract of strawberry leaf and biosynthesized NPs, FT-IR studies were performed on BRUKER FT-IR Spectrophotometer (Billerica, MA, USA). The spectra recorded in the range of 400 to 4000 cm^−1^ (mid-infrared region, MIR) and under the controlled atmospheric conditions.

X-Ray Diffraction (XRD) analysis. XRD analysis was applied to confirm the presence of synthesized NPs, identify the crystal structure, calculate the average particle size and detect the impurities. GNR XRD Explorer instrument (Novara, Italy) with cupper tube, and 40 kv, 30 Ma, 1.5460 Å set up used for x-ray diffraction analysis. The crystalline domain size was calculated using Scherrer equation, Equation (2) [13]:D = κλ/β cos θ(2)
in which θ represents for diffraction angle in radians, D is crystalline domain size and θ is the width of the peak at half of its height in radians. The Scherrer constant, κ, is the shape factor and typically considered to be 0.9. The X-ray wavelength (λ) depends on the type of X-rays used in the experiment (K*α* = 1.5460 Å).

## 4. Conclusions

As the green synthesis of NPs is an ecofriendly, economic and safe method as compared to other synthesis methods, we used a water mediated extraction method with no chemicals for extraction of biomolecules from strawberry leaf. During the biosynthesis of metal-based NPs, the leaf extract added to metal precursors in procedures with different details. Biomolecules presented in strawberry leaf extract, after complexation with metal ions reduce metal ions and then function as the stabilizing or capping agent. Formation of NPs primarily was demonstrated by the intense color change of the reaction mixture and then by UV-Visible spectroscopy for all of the biosynthesized samples. After confirming the biosynthesis of nanoparticles by UV-Vis spectroscopy, the structural properties of the NPs such as size, size distribution, shape, shape heterogeneity and aggregation of them investigated by FESEM. The results confirmed the nanostructure behavior of the synthesized particles. EDS analysis obtained information about the elemental composition of biosynthesized NPs. PCCS analysis gave the average hydrated particle size distribution of the biosynthesized NPs. The FT-IR spectra applied to recognize the functional organic groups or biomolecule residues along with biosynthesized NPs. The obtained FT-IR spectra confirmed the interaction of biomolecules with synthesized NPs. Using XRD studies we confirmed the presence of the NPs, obtained the crystallinity or amorphic nature, detected possible impurities and calculate the average crystallite size of the NPs.

As in this investigation a green, sustainable and highly benign method is introduced, a wide variety of potential applications could be explored for such biosynthesized metal-based nanoparticles, especially in the fields of medicine, drug delivery, biotechnology, catalysis, agriculture and so on. For example, we evaluated the antibacterial activity of biosyntheized silver, copper and ZnO nanoparticles on *Pseudomonas aeruginosa* bacterium and antifungal activity on *Botrytis cinerea* and some other pathogenic fungi; we also studied the effect of these seven biosynthesized nanoparticles on seed germination and seedling growth of wheat and flax seeds which will be published in our future reports.

## Figures and Tables

**Figure 1 molecules-26-03025-f001:**
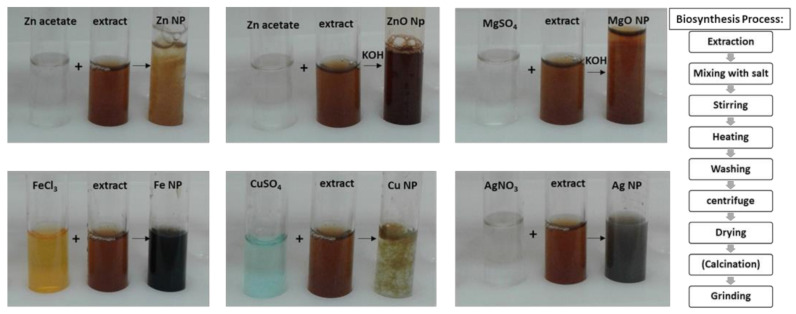
Schematic representation of biosynthesis of NPs and color changes due to the formation of nanoparticles.

**Figure 2 molecules-26-03025-f002:**
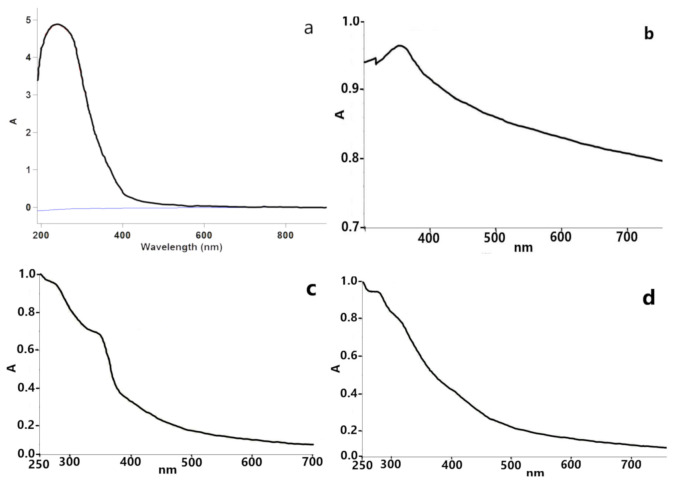
UV-Vis spectra of the (**a**) extract, (**b**) C-ZnO NPs, (**c**) NC-ZnO NP, (**d**) Zn NPs, (**e**) MgO NPs, (**f**) Ag NPs, (**g**) Cu NPs, (**h**) Fe NPs.

**Figure 3 molecules-26-03025-f003:**
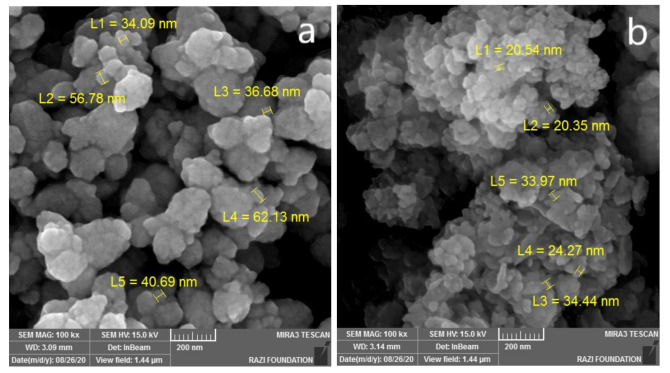
FESEM images of synthesized (**a**) C-ZnO NPs, (**b**) NC-ZnO NPs, (**c**) Zn NPs, (**d**) MgO NPs, (**e**) Ag NPs, (**f**) Cu NPs, (**g**) Fe NPs.

**Figure 4 molecules-26-03025-f004:**
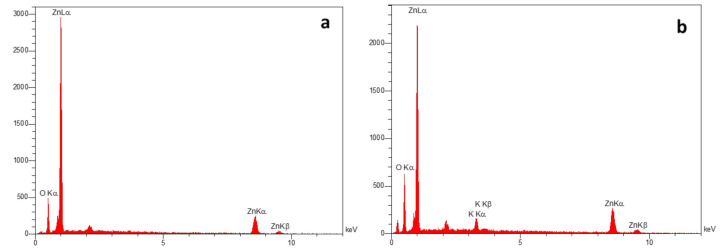
EDS spectrum of synthesized (**a**) C-ZnO NPs, (**b**) NC-ZnO NPs, (**c**) Zn NPs, (**d**) MgO NPs, (**e**) Ag NPs, (**f**) Cu NPs, (**g**) Fe NPs.

**Figure 5 molecules-26-03025-f005:**
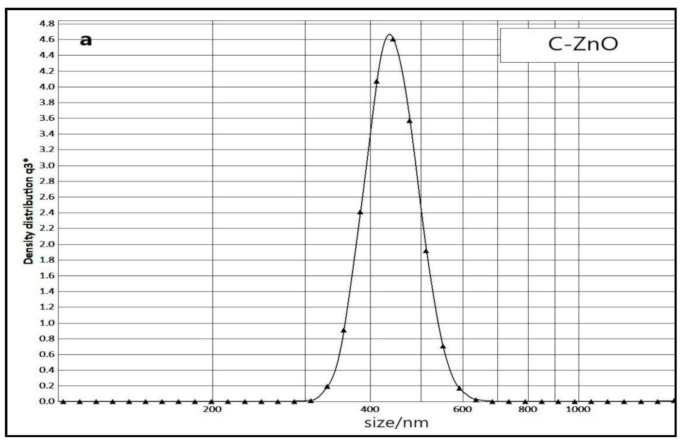
Particle size distribution of (**a**) C-ZnO NPs, (**b**) C-ZnO NPs, (**c**) Zn NPs, (**d**) MgO NPs, (**e**) Ag NPs, (**f**) Cu NPs, (**g**) Fe NPs.

**Figure 6 molecules-26-03025-f006:**
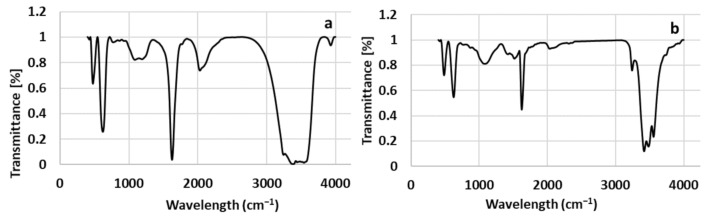
FT−IR spectrum of (**a**) leaf extract, (**b**) C-ZnO NPs, (**c**) NC-ZnO NPs, (**d**) Zn NPs, (**e**) MgO NPs, (**f**) Ag NPs, (**g**) Cu NPs, (**h**) Fe NPs.

**Figure 7 molecules-26-03025-f007:**
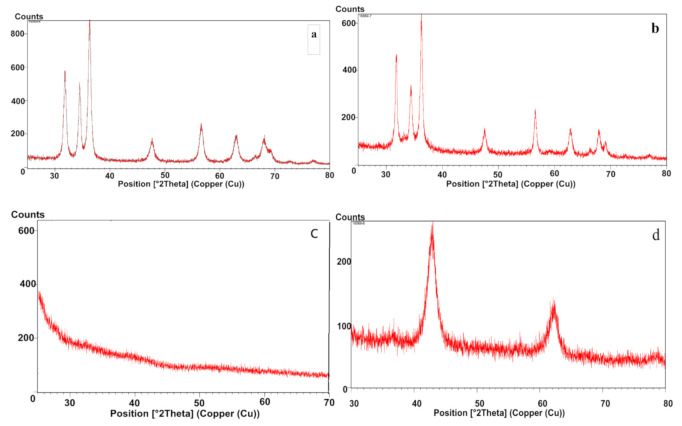
XRD pattern of (**a**) C-ZnO NPs, (**b**) NC-ZnO NPs, (**c**) Zn NPs, (**d**) MgO NPs, (**e**) Ag NPs, (**f**) Cu NPs, (**g**) Fe NPs.

**Table 1 molecules-26-03025-t001:** Average size and shape of the synthesized nanoparticles.

NP	C-ZnO	NC-ZnO	Zn	MgO	Ag	Cu	Fe
Shape	Spherical	spherical	Small sheets	Semi-spherical	spherical	sheets	Small sheets
Diameter (nm)	40	25	100	65	50	180	130
Thickness	-	-	25	-	-	30	20
Standard deviation	9	5	24	36	4	51	29
Homogeneity	yes	yes	no	yes	yes	no	no

**Table 2 molecules-26-03025-t002:** The average hydrated particle size of biosynthesized NPs and the particle size obtained from FESEM, SD: standard deviation, Ave: average hydrodynamic particle diameter of the agglomerates, PDI: polydispersity index, polymodal distribution of NPs.

NP	C-ZnO	NC-ZnO	Zn	MgO	Ag	Cu	Fe
Ave size (nm)	995	210	525	1551	133	795	559
Ave size (nm) of FESEM	40	25	100	65	50	30 × 180	20 × 130
SD	1026	15.16	37.84	278.9	4.47	59.50	31.53
PDI	1.06	0.005	0.005	0.32	0.001	0.005	0.003
Size distribution condition	High polydipersity	Good monodispersity	Good monodispersity	Mid-range monodispersity	Good monodispersity	Good monodispersity	Good monodispersity

**Table 3 molecules-26-03025-t003:** The absorption peaks of extract and synthesized NPs as obtained from FT-IR spectrophotometer and their corresponding functional groups.

Absorption Peak Position (cm^−1^)								Functional Group
Extract	C-ZnO	NC-ZnO	Zn	MgO	Ag	Cu	Fe	
3539	3553	3553	3553	3554	3552	3553	3523	O-H alcohol and phenol N-H primary amine
-	3480	3475	3479	3482	3479	3479	3479	
-	3415	3411	3416	3417	3416	3416	3416	
3392	-	-	-	-	-	-	-	N-H aliphatic primary amine
-	3238	3237	3239	3233	3239	3239	3239	C-H alkene
-	-	-	-	-	-	2923	2922	aliphatic C-H/secondary amine/amine salt
-	-	-	-	2362	-	-	-	CO_2_
2028	2030	2028	2039	1996	2090	2039	2039	C-H aromatic
1624	1621	1620	1622	1621	1622	1622	1622	C=O/C=C aromatic ring
-	1513	-	-	1549	-	-	-	N-O nitro/C=C N-H aromatic amine
-	1424	-	-	-	-	-	-	
-	-	1395	-	-	-	-	-	Sulphate S=O
-	-	-	-	-	-	1327	-	
1084	1079	1069	1148	1078	1151	1151	1144	C-O/C-N amine/C-OH bending
-	-	-	-	871	-	-	-	C-H Alkane/alkene/aromatics/ C-S thiol or thioether
770	-	-	-	-	-	-	-	
621	623	623	621	626	621	621	622	C-H alkyne/O-H alcohol (out of plane bend)
474	481	482	480	486	479	479	480	Organometalics/ S-S sulphides (stretch)
423	418	418	418	-	418	-	418	

## Data Availability

The data presented in this study are available in article.

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
