# Peer review of "Facile Biogenic Synthesis and Characterization of Seven Metal-Based Nanoparticles Conjugated with Phytochemical Bioactives Using Fragaria ananassa Leaf Extract"

_molecules, 2021, doi:10.3390/molecules26103025_

Round 1

Reviewer 1 Report

Authors used Fragaria ananassa (strawberry) leaf extract as a source of natural reducing, capping or stabilizing agents to develop an eco-friendly, cost effective and safe process for the biosynthesis of metal-based nanoparticles. It is not explained because Authors realized these NPs.

It is not reported the potential application of such these NPs both in Introduction and in Conclusions paragraphs. Infact, Authors affirm that a wide variety of potential applications could be explored for such biosynthesized metal based nanoparticles, especially in the fields of medicine, drug delivery, biotechnology, catalysis, agriculture and so on, without to specify a possible precise application for their NSPs. Moreover, although the green synthesis of NPs reported is an ecofriendly, economic and safe method as compared to other synthesis methods, it is not demonstrated if these new synthesized NPs have other advantages. 

A nanoparticle is a small particle that ranges between 1 to 100 nanometres in size. The definition given by the European Commission states that the particle size of at least half of the particles in the number size distribution must measure 100 nm or below. As reported in Table 1, Cu NPs and Fe NPs show a diameter, respectively of 180 and 130 nm. Can they be considered NPs?

Authors should report a distribution of NPs as a function of their size.

Among the keywords it is reported "antibacterial activity" but in the manuscript
there are no experiments for the evaluation of the antibacterial activity.

Table 1 and 2: how is standard deviation? How many NPs have been analysed for each type?

Figure 2: NPs resulted aggregated. It is difficult to measure their sizes accurately

How samples for Field emission scanning electron microscopy (FESEM) with energy dispersive X-ray 507 spectroscopy (EDS) analysis were prepared?

It should be perfomed the same analysis on commercial NPs in order to have a control sample.

Author Response

Dear Reviewer

We gratefully acknowledge the detailed revision of the text and useful suggestions to improve the paper by the reviewer. We have closely followed he/she suggestions and introduced the required changes in the text. Main changes are highlighted into the manuscript in YELLOW. Below, we have included reviewer comments and our responses.

  • The first comment is already addressed in the text of the manuscript.

  • The suggested information added to the introduction and conclusion and highlighted in the text. Moreover, we evaluated the possible applications of these synthesized nanoparticles, but as we obtained a big amount of data, it was not possible to insert them in this paper so we separated the application part in other articles entitled: 1- Study of positive and negative effects of seven biogenic metal-based nanoparticles on seed germination and seedling growth of wheat and flax (submitted); 2- In-vitro evaluation of antibacterial and antifungal activity of biogenic silver and copper nanoparticles (submitted).

Antibacterial activity of C-ZnO and NC-ZnO also evaluated on Pseudomonas aeruginosa and antifungal activity tested on Botrytis cinerea.

  • Advantages of new synthesized NPs adde to the introduction part as suggested by the reviewer.
  • In most of the definitions, nanomaterials are particles with at least one dimension in the nanometer range [C. N. R. Rao, A. Mu¨ller, A. K. Cheetham, The Chemistry of Nanomaterials, 2004, WILEY-VCH Verlag GmbH & Co. KGaA, Weinheim, ISBN: 3-527-30686-2]. Cu and Fe nanoparticles have at least one dimension in the range of nanometer and could be considered as nanomaterials. Two dimensions allocated for Cu, Fe and Zn NPs in table 1 (diameter and thickness) because these NPs are in form of sheets and diameter and thickness can describe the shape of these NPs. Cu has the thickness of 30 and Fe 20nm.

Also in some definitions NPs could be as large as 100 to 200nm [Zhong Lin Wang, characterization of nanophase materials, 2000, WILEY-VCH Verlag GmbH & Co.].

  • Figure 4 represents particle size distribution of synthesized NPs.

  • Keywords revised.

  • SD added to the Table 1 and number of the particles added to the materials and methods part-highlighted.

For table 2, The spectroscopy was repeated 5 times for each NP and the average graph was considered for preparing the table. In the table 2 the standard deviation is represented in the third row as SD, which was automatically calculated by instrument’s software. 

  • Using “Image J software” is helpful in measuring such cases. Moreover, the calculated size from XRD spectrum was in accordance with the size calculated from FESEM images.

  • Preparation process added to the materials and methods part.

  • About commercial NPs, before designing our experiment, we did not face with such comparisons in literature, but it will be a good idea to compare the results with a commercial product.

We hope that after these enhancements the manuscript can now be accepted, although we are certainly willing to consider further changes if necessary.

Yours sincerely,

Reviewer 2 Report

This is an interesting manuscript, however, I think that if the Authors included in this paper some biological examination of the synthesized nanoparticles the manuscript will be more interesting and will generate higher scientific soundness. 

Major comments:

1. In the photography of morphology of NPs some aggregates has been indicated, so may addition of other techniques such as TEM will provide more information about their structural morphology and will be more helpful in recognition of NPs size 

2. The FTIR spectra might be analyzed in Far-IR set-up to get some information about metal-metal bonds - it can be useful in the cases of metallic NPs analysis

3.The including of some additionall biological experiment to identify their properties is suggested.

Minior

Please improve the figures quality

Some schematic illustration of synthesis process will enhance the quality of this study. 

Author Response

Dear Reviewer

We gratefully acknowledge the detailed revision of the text and useful suggestions to improve the paper by the reviewer. We have closely followed he/she suggestions and introduced the required changes in the text. Main changes are highlighted into the manuscript in YELLOW. Below, we have included reviewer comments and our responses.

  • That is true, the presented FESEM images show aggregates but it was not difficult to measure the particle size using “image J software”. Moreover, using this software, the borders of the individual particles are clear enough to be measured.

  • As we designed our research upon previously published literature, FTIR spectra in the range of 400-4000 cm-1 was used for confirming the presence of phytochemicals on the surface of biosynthesized nanoparticles and we did not face with Far-IR set-up. Also the peak shifts in comparison with extract could be related to the metal-ligand interactions. There are single metals participated in each synthesized nanoparticles and metal-metal interactions could not be studied, but it is a valuable suggestion to use Far-IR set-up to study metal-ligand bonds.

  • We evaluated the possible applications of these synthesized nanoparticles, but as we obtained a large amount of data, it was not possible to insert them in this paper so we separated the application part in other articles entitled: 1- Study of positive and negative effects of seven biogenic metal-based nanoparticles on seed germination and seedling growth of wheat and flax (submitted); 2- In-vitro evaluation of antibacterial and antifungal activity of biogenic silver and copper nanoparticles (submitted).

Antibacterial activity of C-ZnO and NC-ZnO also evaluated on Pseudomonas aeruginosa and antifungal activity tested on Botrytis cinerea.

  • All figures revised as it was suggested.

  • Some schematic illustration of synthesis process added to the manuscript.

We hope that after these enhancements the manuscript can now be accepted, although we are certainly willing to consider further changes if necessary.

Yours sincerely,

Round 2

Reviewer 1 Report

Authors have been improved manuscript as suggested and now it is suitable for publication.

Reviewer 2 Report

The authors responded to my queries at a satisfactory level.